# Superfluids as higher-form anomalies

**Luca V. Delacrétaz[1], Diego M. Hofman[2] and Grégoire Mathys[2]**

**1** Stanford Institute for Theoretical Physics, Stanford University, Stanford, California, USA
**2** Institute for Theoretical Physics, University of Amsterdam, Amsterdam, The Netherlands

## Abstract

We recast superfluid hydrodynamics as the hydrodynamic theory of a system with an emergent anomalous higher-form symmetry. The higher-form charge counts the winding planes of the superfluid – its constitutive relation replaces the Josephson relation of conventional superfluid hydrodynamics. This formulation puts all hydrodynamic equations on equal footing. The anomalous Ward identity can be used as an alternative starting point to prove the existence of a Goldstone boson, without reference to spontaneous symmetry breaking. This provides an alternative characterization of Landau phase transitions in terms of higher-form symmetries and their anomalies instead of how the symmetries are realized. This treatment is more general and, in particular, includes the case of BKT transitions. As an application of this formalism we construct the hydrodynamic theories of conventional (0-form) and 1-form superfluids.

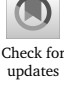

# 1  Preliminaries and framework

Consider a superfluid, i.e. a system that spontaneously breaks a $U(1)$ symmetry. If it is Lorentz invariant, it can be described by the low energy effective field theory (EFT) [1, 2]

$$S = \int d^d x \, P\left(\sqrt{-D_\mu \phi D^\mu \phi}\right) + \cdots, \tag{1.1}$$

with $D_\mu \phi = \partial_\mu \phi - q A_\mu$. Here, $P(\cdot)$ is a smooth function away from zero and the ellipses denote higher derivative terms. From the high energy perspective, $\phi$ represents the phase of the charged operator that condenses, $q$ is its charge and $A_\mu$ is a background $U(1)$ gauge field[1]. The $U(1)$ current is given by

$$J_\mu = \frac{\delta S}{\delta(D_\mu \phi)} = P' \frac{D_\mu \phi}{\sqrt{-D_\nu \phi D^\nu \phi}} + \cdots. \tag{1.2}$$

Now, following the nomenclature from [3], notice that the EFT also enjoys a $(d-2)$-form symmetry $U(1)^{(d-2)}$ carried by the current $K_{\mu_1 \mu_2 \ldots \mu_{d-1}}$ given, compactly, by[2]

$$(\star K)_\mu = D_\mu \phi, \tag{1.3}$$

where $\star$ is the Hodge dual operator. Charged objects under this symmetry are winding planes of the superfluid phase $\phi$. This higher form symmetry is explicitly broken by the proliferation of vortices as one returns to the normal phase, and typically will not be a symmetry of the microscopic theory: it is an emergent symmetry of the superfluid phase. In the presence of non-trivial background gauge fields $F = dA \neq 0$, the conservation of the higher form current is also broken at low energies by an anomaly as

$$d \star K = -aF, \tag{1.4}$$

where $a$ is the anomaly coefficient. It can be connected to UV data by $a = q$. Notice that even without invoking UV arguments relating to charge quantization, it is easy to see that flux quantization implies $a \in \mathbb{Z}$ for a compact $U(1)$ symmetry,

Unlike axial-type anomalies, this mixed anomaly between $U(1)$ and $U(1)^{(d-2)}$ symmetries can occur in any dimension. It is similar to the axial anomaly in $d = 2$ (e.g. in the Schwinger model) and generalizes it to higher dimensions. This anomaly has a simple physical interpretation: without background fields, the number of winding planes (or the supercurrent in a superconductor) is conserved. In an external electric field the number of winding planes (or the supercurrent) will increase linearly in time. Winding planes in any direction can be added or removed by turning on an appropriate electric field.

---

[1]The ultraviolet reader uncomfortable with this EFT will find a discussion in section 1.2 on how this theory emerges as the low-energy description of certain microscopic models.

[2]The current associated with the higher-form symmetry is $(\star K)_\mu = \partial_\mu \phi$. Nevertheless, this current is not invariant under $U(1)$ gauge transformations, and the gauge invariant combination is given in (1.3), which is not conserved. The non-conservation of (1.3) leads to the anomalous conservation equation (1.4).

Spontaneous symmetry breaking (SSB) leads, therefore, to an emergent $(d-2)$-form symmetry with anomaly (1.4). In section 1.1 below, we show that there exists an (almost) converse statement, namely a system with $U(1) \times U(1)^{(d-2)}$ symmetry with anomaly (1.4) contains a massless Goldstone boson transforming non-linearly in its spectrum[3]. As a consequence, SSB phases of systems enjoying abelian symmetries can equivalently be formulated in terms of mixed anomalies (1.4). It is tempting to adjust Landau's paradigm for classifying phases by only specifying *which* generalized symmetries each phase has, along with their anomalies, disregarding *how* they are realized – linearly or non-linearly (i.e. whether the symmetries are spontaneously broken or not). For example, the BKT transition in $2+1$ dimensional superfluids is sometimes said to be non-Landau because there is only quasi-long range order at low but finite temperatures. Generalized symmetries however distinguish both phases. Conservation of the emergent higher form symmetry in the superfluid phase has tangible consequences: it leads, in particular, to an infinite dc conductivity $\sigma(\omega) \sim i/\omega$, observed experimentally even in $2+1$ dimensions e.g. in thin superconducting films or superfluids when $T < T_{\text{BKT}}$ [4, 5].

The philosophy of insisting on symmetries alone rather than how they are realized on microscopic fields plays a central role in hydrodynamics. In this paper, we propose to recast superfluid hydrodynamics as a the hydrodynamical theory of a system with $U(1) \times U(1)^{(d-2)}$ symmetry with a mixed anomaly (1.4). This fomulation puts all hydrodynamic equations on equal footing – as conservation laws and constitutive relations for the various currents. The 'Josephson relation' in the standard treatment of superfluid hydrodynamics [6] is replaced by the constitutive relation for the higher form current (see Ref. [7] for the analogous statement in the context of spontaneous breaking of translation symmetry). This streamlines the hydrodynamics algorithm along the lines of what was done recently with magnetohydrodynamics [8]. How anomalies enter hydrodynamics has been understood since the seminal work of Son and Surowka [9] where it was shown that the chiral anomaly fixes terms in the constitutive relations at first order in derivatives. Furthermore, the understanding of the interplay between hydrodynamics and anomalies has led to many important results in the field, e.g. see [10–17]. The case at hand is in some sense the simplest anomaly in hydrodynamics, since it enters at zeroth order in derivatives.

Mixed anomalies of higher form symmetries have also been discussed recently in the context of 2-groups [18,19] and for discrete symmetries in the context of topological phases [20].

## 1.1 An alternative to Goldstone's theorem

The standard input for the Nambu-Goldstone theorem is that a symmetry breaking order parameter acquires a vacuum expectation value. Here, we obtain the equivalent result for relativistic QFTs with a different starting point; namely that the theory has a global symmetry $U(1) \times U(1)^{(d-2)}$ with mixed anomaly

$$\partial_\mu \langle J^\mu \rangle = 0, \qquad \partial_{[\mu} \langle (\star K)_{\nu]} \rangle = -a\, F_{\mu\nu}. \tag{1.5}$$

The Fourier transform of the mixed correlator is constrained by Lorentz invariance to take the form

$$\Pi_{\mu\nu}(p) \equiv \int d^d x\, e^{ixp} \left\langle \mathcal{T}(\star K)_\mu(x) J_\nu(0) \right\rangle = f(p^2) p_\mu p_\nu + g(p^2) p^2 g_{\mu\nu}, \tag{1.6}$$

where $\mathcal{T}$ denotes time-ordering. The Ward identity for the 0-form current gives

$$\Pi_{\mu\nu} p^\nu = 0 \qquad \Rightarrow \qquad f(p^2) + g(p^2) = 0. \tag{1.7}$$

---

[3]Strictly speaking, this is weaker than SSB as there does not need to be a charged operator that acquires an expectation value. Therefore, the symmetry structure presented here, including the anomaly, is a weaker assumption than SSB, making it more general.

The anomalous Ward identity for the $(d-2)$-form current (1.5) reads

$$p_{[\alpha}\Pi_{\mu]\nu} = -a\, p_{[\alpha}g_{\mu]\nu} \qquad \Rightarrow \qquad g(p^2) = -\frac{a}{p^2}\,. \tag{1.8}$$

The mixed correlator is therefore completely fixed by the anomaly

$$\Pi_{\mu\nu}(p) = a\,\frac{p_\mu p_\nu - p^2 g_{\mu\nu}}{p^2}\,. \tag{1.9}$$

An important remark here is that the $U(1)^{(d-2)}$ symmetry is emergent, and is broken by vortices. This will lead to corrections to (1.9) which are non-singular as $p^2 \to 0$ and vanish when $p_\mu \to 0$, since the vortices are gapped. The $p^2 = 0$ pole in (1.9) is therefore a robust consequence of the anomaly (1.5) of the emergent symmetry, and is all that is needed for this proof.

We now proceed along lines similar to current-algebra proofs of Goldstone's theorem [1]. The Källén-Lehmann representation of the time-ordered correlator (1.9) is[4]

$$\Pi_{\mu\nu}(p) = \int_0^\infty d\mu^2\, \rho_{KJ}(\mu^2)\frac{p_\mu p_\nu - p^2 g_{\mu\nu}}{p^2 - \mu^2 + i\epsilon}\,, \tag{1.10}$$

where the spectral density $\rho_{KJ}(p^2)$ (non-sign definite since it involves two different operators) is defined as

$$\sum_n (2\pi)^d \delta^d(p - p_n)\langle 0|(\star K)_\mu(0)|n\rangle\langle 0|J_\nu(0)|n\rangle^* \equiv \rho_{KJ}(p^2)p_\mu p_\nu\,. \tag{1.11}$$

Comparing with (1.9) one immediately concludes that there exists a massless state $p_n^2 = 0$ that is created by both currents, i.e.

$$\rho_{KJ}(\mu^2) = a\,\delta(\mu^2) + \cdots\,, \tag{1.12}$$

where $\cdots$ are contributions that are finite as $\mu \to 0$.

Notice that, up to contact terms (see footnote 4) and identifying $(\star K_\mu) = \partial_\mu \phi$, we can interpret (1.9) as coming from the momentum space correlation function

$$\langle \phi\, J_\nu \rangle = a\frac{p_\nu}{p^2}\,. \tag{1.13}$$

This precisely satisfies the Ward identity for a field $\phi$ transforming non-linearly under the $U(1)$ induced by $J_\mu$, indicating that this symmetry is spontaneously broken in $d > 2$. While there is no spontaneous symmetry breaking in $d \leq 2$, our arguments do go through even in that case showing that our setup is more general than the usual classification of phases by the realization of symmetries and includes more exotic cases, such as BKT transitions.

A short but important conclusion from this analysis is that it is really anomalies that are responsible for the existence of massless modes, as a more general statement than symmetry breaking. This discussion connects with the study of topological phases [22, 23], where topological insulators accommodate massless modes at their boundaries, stabilized by anomaly inflow. The present anomaly (1.4) can be canceled by inflow from a bulk with the term $S_{\text{bulk}} = a\int B \wedge F$, where $B$ is the $(d-1)$-form source for the current $K$.

A mixed anomaly can similarly be seen to protect the masslessness of the photon or higher-form gauge fields – the hydrodynamics of such a system is discussed in Sec. 4. The proof above

---

[4] A non-covariant contact term has to be added to make the time-ordered correlator of spin-1 operators covariant. This can be done while preserving Ward identities, see [21].

can easily be generalized to the case where an anomalous $U(1)^{(p)} \times U(1)^{(d-p-2)}$ is present, leading to the presence of $p$-form massless gauge fields in the spectrum for $0 \leq p \leq d-2$. When $d = 2p+2$, we expect the emergence of a conformal phase at low energies, as the gauge coupling constant is dimensionless. In this very special case the converse of this statement was proven in [24]: a conformal theory enjoying a $U(1)^{(p)}$ symmetry in $d = 2p+2$ dimensions must also have an anomalous $U(1)^{(d-p-2)}$.

## 1.2 Scales and defects in the superfluid EFT

Before studying the hydrodynamics we pause to make a few comments on the effective field theory (1.1). A paradigmatic microscopic model that leads to it is the Landau-Ginzburg model for a complex scalar

$$\mathcal{L} = -\frac{1}{2}|D_\mu \Phi|^2 - V(|\Phi|^2),\qquad(1.14)$$

where $D_\mu \Phi \equiv \partial_\mu \Phi + i A_\mu \Phi$ and with potential

$$V(\rho^2) = -\frac{1}{2}m^2 \rho^2 + \frac{g}{4}\rho^4 = \frac{g}{4}(\rho^2 - v^2)^2 + \text{const},\qquad(1.15)$$

where $v = m/\sqrt{g}$. Expanding around the saddle $\Phi = (v + r)e^{i\phi}$ and integrating out the radial mode at tree level this leads to (for energies $E \ll m$)

$$S_{\text{eff}} = -v^2 \int d^d x \frac{1}{2}(D_\mu \phi)^2 + \frac{a_4}{m^2}(D_\mu \phi)^4 + \frac{a_6}{m^4}(D_\mu \phi)^6 + \cdots,\qquad(1.16)$$

where the $a_n$ are combinatorial factors. This is clearly a special case of (1.1), where[5] we can understand $P(\mu)$ as an analytic series expansion in $\mu^2$. The strong coupling scale in this model is $\Lambda_{\text{sc}} \sim \left(\frac{m^4}{g}\right)^{\frac{1}{d}}$, which, as usual, is parametrically larger than the scale of new physics $m$ if the UV is weakly coupled. In this cases we can imagine resumming the series to obtain a function $P(\mu)$ and still retain perturbative control. The leads to the treatment discussed around (1.1).

From the point of view of the higher form current (1.3), the UV scale $\Lambda_{sc}$ has clear physical meaning. The theory (1.14) admits vortex solutions which can be constructed (up to logarithmically IR divergent terms) as soon as we hit the symmetry restoration scale given by $\Lambda_{\text{sc}}$. When this happens, the winding planes charged under $U(1)^{(d-2)}$ can end on the vortices and the symmetry becomes explicitly broken.

Of course this weakly coupled description does not have to be valid. While the superfluid system described above exists even with zero chemical potential, so that one can consistently take $\mu \ll \Lambda_{\text{sc}}$, certain superfluid phases only occur at finite chemical potential (such as in QCD). In this case the strong coupling scale is typically of order the chemical potential. A simple example of such a situation is a conformally invariant superfluid, where $P(\mu) = \alpha \mu^d$ by scale invariance so $\Lambda_{\text{sc}} \sim \mu$ (in this case $P(\mu)$ might not be analytic in $\mu^2$). More generally the equation of state $P(\mu)$ entirely fixes the EFT at leading order in gradients – the equation may however be complicated away from the conformal situation. See [25] for an extended discussion.

## 1.3 Decoupling of currents in the superfluid EFT

The reader may wonder to what extent the two currents (1.2) and (1.3) should be treated as independent vectors in the hydrodynamic description. Although the operators $J_\mu$ and $(\star K)_\mu$ are

---

[5]This notation is natural if one notices that for constant field configurations, in a background time-like field, the argument of $P(\mu)$ is indeed the chemical potential.

different (for example in (1.16) they differ by terms suppressed in $m^2$), they remain collinear when evaluated in any given field configuration. This is no longer the case in a thermal ensemble. $T \neq 0$ introduces a preferred vector ($u^\mu = \delta_0^\mu$ in the rest frame of the fluid) which together with the superfluid winding distinguishes both currents. Since $u_\mu$ is even under charge conjugation[6], the decoupling of currents can only happen when also at finite chemical potential $\mu \neq 0$. The difference between the currents corresponds to the 'normal density' in the two-fluid picture of finite temperature superfluids, coming from thermally populated superfluid phonons. In this section we show how a thermal 1-loop computation in the effective theory (1.1) distinguishes the two currents at finite (but small) temperature $T$. Although this calculation has been done in certain microscopic models that exhibit superfluidity (see e.g. [26] for a field theory calculation in the Landau-Ginzburg model), we are not aware of a calculation in the universal EFT. This decoupling of the currents at finite temperature justifies why they should be treated as independent vectors in the hydrodynamic setup of section 3.

Taking $A_\mu = \mu \delta_\mu^0$ and expanding the action in fields around a solution with finite superfluid winding[7] $\partial_\mu \phi \to \delta_\mu^I \tilde{\rho}_I + \partial_\mu \phi$ leads to

$$
\begin{aligned}
S &= \int d^d x \, P(\sqrt{-D_\mu \phi D^\mu \phi}) + \cdots \\
&= \int d^d x \, P - \frac{P'}{2X_0} (\partial_\mu \phi)^2 + \frac{1}{2} \left( \frac{P''}{X_0^2} - \frac{P'}{X_0^3} \right) (\mu \dot{\phi} + \tilde{\rho} \cdot \nabla \phi)^2 + O(\partial \phi)^3 + \cdots,
\end{aligned}
\tag{1.17}
$$

where the functions without argument ($P, P'$, etc.) are evaluated at $X_0 \equiv \sqrt{\mu^2 - \tilde{\rho}^2}$. We can identify these with the pressure $P$, charge density $\rho = P'$ and susceptibility $\chi = P''$ at zero temperature. The speed of superfluid sound is clearly anisotropic, see [26] for an extended discussion.

The currents in this theory are given by (1.2) and (1.3); expanding again in fields gives

$$
(\star K)_\mu = \partial_\mu \phi + \delta_\mu^I \tilde{\rho}_I - \mu \delta_\mu^0,
\tag{1.18}
$$

$$
J_\mu = \frac{(\star K)_\mu}{X_0} \left[ P' - \left( \frac{P''}{X_0} - \frac{P'}{X_0^2} \right) (\mu \dot{\phi} + \tilde{\rho} \cdot \nabla \phi) + O(\partial \phi)^2 + \cdots \right].
\tag{1.19}
$$

In the ground state the normal ordered operators above have no expectation value and we have the densities $\langle J_0 \rangle = P' = \rho$ and $\langle (\star K)_I \rangle = \tilde{\rho}_I$ at $T = 0$ as expected. For any single field configuration, the two currents are manifestly parallel. However they are distinguished in the finite temperature ensemble, where $\dot{\phi}^2$ and $\nabla \phi^2$ acquire thermal expectation values at 1-loop. Here we will work at small temperature for simplicity so that the equation of state can be expanded around $T = 0$. The expectation value of the dual current is simply

$$
\langle (\star K)_\mu \rangle_\beta = \delta_\mu^I \tilde{\rho}_I - \mu \delta_\mu^0.
\tag{1.20}
$$

The finite temperature correction in the direction of the regular current is

$$
\langle J_\mu \rangle_\beta - \langle J_\mu \rangle = - \left( \frac{P''}{X_0^2} - \frac{P'}{X_0^3} \right) \langle \partial_\mu \phi (\mu \dot{\phi} + \tilde{\rho} \cdot \nabla \phi) \rangle_\beta.
\tag{1.21}
$$

Here we neglected two contributions to $\langle J_\mu \rangle_\beta$, coming from the temperature dependence of $P'$ and the thermal expectation value of the $O(\partial \phi)^2$ term in (1.19). Both of these will give corrections to the magnitude of $\langle J_\mu \rangle_\beta$, but not to its direction which we are interested in. In the

---

[6]Charge conjugation acts in the EFT (1.1) as $\phi \to -\phi, \mu \to -\mu$.

[7]The index $I = 1, 2, \ldots, d-1$ runs over the spatial dimensions.

traditional language, they are finite temperature corrections to the superfluid density, instead of contributions to the normal density.

In order to prove that the two currents are independent, one must compute the thermal expectation value (1.21) in the linearized theory (1.17) and show that it is not parallel to (1.20). We will do so to leading order in $\tilde{\rho}$.

$$\langle \partial_\mu \phi (\mu \dot{\phi} + \tilde{\rho} \cdot \nabla \phi) \rangle_\beta = \left( \mu \delta_\mu^0 \langle \dot{\phi}^2 \rangle_\beta + \frac{1}{d-1} \delta_\mu^I \tilde{\rho}_I \langle \nabla \phi^2 \rangle_\beta + \mu \delta_\mu^I \langle \dot{\phi} \partial_I \phi \rangle_\beta \right) + O(\tilde{\rho}^2). \quad (1.22)$$

The first two terms can be computed in the isotropic theory, where a 1-loop calculation with appropriate UV regulator gives

$$\langle \dot{\phi}^2 \rangle_\beta = c_s^2 \langle \nabla \phi^2 \rangle_\beta = \frac{(d-1) f_d}{\beta^d c_s^{d-1} P''}, \qquad \text{with} \quad f_d = \frac{\Gamma(\frac{d}{2}) \zeta(d)}{\pi^{d/2}}, \quad (1.23)$$

where the isotropic speed of sound is given by $c_s^2 = \frac{P'}{\mu P''}$. The last term is slightly more subtle but can be computed similarly, one finds

$$\langle \dot{\phi} \partial_I \phi \rangle_\beta = \frac{1 - c_s^2}{c_s^2} \frac{\tilde{\rho}_I}{\mu} \langle \dot{\phi}^2 \rangle_\beta + O(\tilde{\rho}^3). \quad (1.24)$$

We therefore find that the contribution (1.21) to the current is

$$\langle J_\mu \rangle_\beta - \langle J_\mu \rangle = \frac{1 - c_s^2}{\mu \beta^d c_s^{d-1}} (d-1) f_d \left[ \delta_\mu^0 + \frac{\tilde{\rho}_I \delta_\mu^I}{\mu} \left( \frac{d}{(d-1) c_s^2} - 1 \right) \right], \quad (1.25)$$

which is never parallel to (1.20). At low temperatures, we see that as long as $c_s^2 < 1$ the deviation between the two currents (and therefore the 'normal density') is $\sim T^d$, in agreement with standard results (see e.g. [27]).

## 2 An incoherent superfluid appetizer

The existence of the anomaly (1.4) applies to any local system with a spontaneously broken $U(1)$ symmetry and does not rely on translational or boost invariance. This leads us to consider a system where conservation of energy and momentum can be ignored[8] and focus on the hydrodynamics of the conserved currents (1.2) and (1.3) alone. This constitutes the simplest instance of a hydrodynamic system with the (anomalous) symmetry structure discussed in the introduction. The XY model on a lattice is a simple microscopic realization of such a system.

The full hydrodynamics, including energy-momentum, is treated in the next section in a systematic manner (including a careful study of the role of anomalies); we take advantage of the simpler setting in this section to make the conceptual issues more clear. We therefore consider the hydrodynamics of a system satisfying the conservation laws

$$d \star J = 0, \quad (2.1)$$

$$d \star K = -aF, \quad (2.2)$$

with $a \in \mathbb{Z}$ where $J$ and $(\star K)$ are one forms and $F = dA$ is a background two-form for the $U(1)$ gauge field that couples to $J$. At finite temperature, there exists a preferred rest frame for this system given by a (non-dynamical) velocity field $u^\mu$. This allows us to discuss the physics in

---

[8]Strictly speaking, one would also have to consider the conservation of energy. For simplicity we disregard this contribution which would just lead to an additional diffusive mode.

manifest $SO(d-1)$ non-relativistic notation in what follows. We will consider the case where there are no background sources, $A = 0$.

The hydrodynamic variables are the charge densities $J^0 = \rho$ and $(\star K)^I = \tilde{\rho}^I$. We will denote their conjugate dynamical potentials by $\mu$ and $\tilde{\mu}^I$ and the corresponding susceptibilities[9] $\chi$ and $\tilde{\chi}$. As a further simplification in this section, we will assume the background dual potential vanishes $\bar{\tilde{\mu}}^I = 0$ (corresponding to the absence of a background winding of the superfluid). This assumption is lifted in the general treatment of Sec. 3.

The most general constitutive relations up to first order in derivatives are

$$J^I = a\tilde{\mu}^I - \sigma \partial^I \mu + \cdots, \tag{2.3}$$

$$(\star K)^0 = a\mu - \tilde{\sigma} \nabla \cdot \tilde{\mu} + \cdots. \tag{2.4}$$

Onsager relations require both terms that are zeroth order in derivatives to have the same coefficient, which is fixed to be the anomaly coefficient $a$ by Luttinger's argument[10]. There are only two transport parameters $\sigma, \tilde{\sigma} \geq 0$, which are positive by the second law of thermodynamics. Identifying temporarily $\star K$ with the gradient of the superfluid phase reproduces the equations of conventional superfluid hydrodynamics, in particular (2.4) gives rise to the Josephson relation (see e.g. Eqs. (11) and (13) in Ref. [29]).

The conservation equations read

$$0 = \chi \partial_t \mu + a \partial_I \tilde{\mu}^I - \sigma \nabla^2 \mu + \dots, \tag{2.5}$$

$$0 = \tilde{\chi} \partial_t \tilde{\mu}^I + a \partial_I \mu - \tilde{\sigma} \nabla^2 \tilde{\mu}^I + \dots, \tag{2.6}$$

$$0 = \partial_I \tilde{\mu}_J - \partial_J \tilde{\mu}_I. \tag{2.7}$$

These equations represent two physical (first order) modes, as the third equation above is a constraint. They combine into a single (second order) damped sound mode

$$\omega = \pm \frac{a}{\sqrt{\chi \tilde{\chi}}} |k| - \frac{i}{2} \left( \frac{\sigma}{\chi} + \frac{\tilde{\sigma}}{\tilde{\chi}} \right) k^2 + O(k^3). \tag{2.8}$$

Introducing a background dual potential $\bar{\tilde{\mu}}^I \neq 0$ would lead to anisotropies in the speed of sound (as was shown in the non-dissipative treatment of section 1.3) and in the sound attenuation rate.

## 2.1 Phase relaxation and vortices

In this language, phase relaxation due to proliferating vortices is naturally captured as explicit breaking of the higher form symmetry. If the explicit breaking is weak (i.e. if the relaxation rate is small in units of the hydrodynamics cutoff), it can be incorporated in the hydrodynamics by replacing the higher form conservation equation with

$$\partial_\mu K^{\mu\nu} = \Gamma u_\mu K^{\mu\nu} + \cdots, \tag{2.9}$$

to leading order in derivatives. Here we specialized to $2+1$ dimensions so that vortices do not break isotropy. As usual with weak explicit breaking of symmetries, the relaxation rate $\Gamma$ can be related to microscopic relaxation mechanisms via a Kubo formula [30]

$$\Gamma \delta_{IJ} = \lim_{\omega \to 0} \frac{1}{\omega} \operatorname{Im} G^R_{\dot{K}_{0I} \dot{K}_{0J}}(\omega). \tag{2.10}$$

See Refs. [29,31] for applications of this Kubo formula to thin film incoherent superconductors. Generalized symmetries therefore allow to recast weak phase relaxation as weak breaking of higher form symmetries.

---

[9]The dual susceptibility is related to the superfluid stiffness $f_s$ as $\tilde{\chi} = 1/f_s$.

[10]The argument [28] is as follows: in equilibrium the charge densities respond to background fields $\delta J_0 = \chi A_0$, and the currents vanish. Their constitutive relations should therefore be functions of $\frac{1}{\chi} \delta J_0 - A_0 = \delta \mu - A_0$.

# 3 Relativistic superfluid hydrodynamics

In this section we will systematically construct the complete hydrodynamics of a system enjoying an (anomalous) $U(1) \times U(1)^{(d-2)}$ symmetry, including its energy-momentum sector, to first non-trivial order in derivatives. We will show that the result agrees precisely with previous results in the literature [32–34], without invoking any extra assumptions except for the symmetries but with no reference to the character of their realization.

The way to construct a hydrodynamic theory is to write down the most general constitutive relations for all conserved quantities in the system in terms of equilibrium thermodynamic functions and the tensor structure that represents the (explicit) breaking of space-time symmetries.

## 3.1 Zeroth-order hydrodynamics

We want to build the hydrodynamical theory for $d$-dimensional relativistic superfluids. Therefore we must have a conserved energy-momentum tensor $T^{\mu\nu}$, a conserved current $J^\mu$ and a second, anomalous, conserved current $K^{\mu_1\cdots\mu_{d-1}}$ that is associated to the dual symmetry. We expect that the system can be completely described in terms of three scalars and two vectors that we take to be the temperature $T$, two chemical potentials $\mu$ and $\tilde\mu$, a velocity vector $u^\mu$ that specifies the rest frame and a vector $h^\mu$ that specifies the orientation of the co-dimension 1 charged objects (i.e. planes) under $K$. Since it is the codimension of these objects that is fixed, it is more convenient to write the hydrodynamics in terms of the Hodge dual $\star K$ instead of $K$. Furthermore we have the freedom to consider orthonormalized vectors as

$$u^\mu u_\mu = -1, \qquad h^\mu h_\mu = 1, \qquad u^\mu h_\mu = 0. \tag{3.1}$$

In addition, we can define a projector onto the plane orthogonal to both $u^\mu$ and $h^\mu$ as

$$\Delta^{\mu\nu} = \eta^{\mu\nu} + u^\mu u^\nu - h^\mu h^\nu, \tag{3.2}$$

whose trace is $\Delta_\mu{}^\mu = d - 2$.

The most general expressions for the conserved tensors in terms of these quantities at zeroth order in derivatives are

$$T^{\mu\nu} = (\epsilon + p - \tau)u^\mu u^\nu + (p - \tau)\eta^{\mu\nu} + \tau h^\mu h^\nu + \gamma u^{(\mu} h^{\nu)}, \tag{3.3}$$

$$J^\mu = \rho u^\mu + \sigma h^\mu, \tag{3.4}$$

$$(\star K)^\mu = \tilde\sigma u^\mu + \tilde\rho h^\mu, \tag{3.5}$$

where all scalar functions are understood to depend on $T$, $\mu$ and $\tilde\mu$. We define symmetrization and antisymmetrization without the conventional factor of two, i.e $u^{(\mu} h^{\nu)} = u^\mu h^\nu + u^\nu h^\mu$.

In the presence of a background field $A_\mu$ for the current $J^\mu$, the conservation equations read

$$\partial_\mu T^{\mu\nu} = F^{\nu\rho} J_\rho, \tag{3.6}$$

$$\partial_\mu J^\mu = 0, \tag{3.7}$$

$$\partial_{[\mu}(\star K)_{\nu]} = -a F_{\mu\nu}, \tag{3.8}$$

where

$$F_{\mu\nu} = \partial_\mu A_\nu - \partial_\nu A_\mu, \tag{3.9}$$

and we have allowed for an anomaly coefficient $a$.

As a sanity check we see that we have $2d$ dynamical equations[11] for $2d$ degrees of freedom contained in $T, \mu, \tilde{\mu}, u^{\mu}, h^{\mu}$. While we expect one equation of state to fix the scalar functions $p, \epsilon, \rho, \tilde{\rho}$ in (3.6-3.8) in terms of $T, \mu, \tilde{\mu}$, the remaining four scalars must be fixed by other means.

First, the function $\gamma$ can be fixed to any desired value by boosting the system in the $(u, h)$ plane. This preserves the norms, so it is an ambiguity of the parametrization. Normally we would like to pick $\gamma = 0$, but we will keep it arbitrary for now as it simplifies the discussion of the anomaly. We will fix it by demanding that the entropy current is at rest in the frame given by $u^{\mu}$ later on.

Second, $\tau$ corresponds to the tension of the charged planes in the fluid. It can be uniquely fixed using a thermodynamic argument equivalent to the one displayed in [8] in the case of magnetohydrodynamics. It amounts to showing that this tension has to be a particular fixed function of $\tilde{\mu}$ and $\tilde{\rho}$ in order for our system to show the thermodynamic volume scaling characteristic of local theories. We reproduce this argument in appendix A, but we will shortly show that this is not necessary as this coefficient is fixed uniquely, as well, by entropy conservation.

Lastly, $\sigma$ and $\tilde{\sigma}$ will be fixed by the anomaly and its effect in the conservation of the entropy current. If there were no anomaly, we would just set $\sigma = \tilde{\sigma} = 0$ as we would consider a frame where the charges are at rest simultaneously with the entropy. Once the anomaly is included we will see this is no longer possible.

This discussion makes the system of equations closed. The thermodynamics is hence completely fixed by a single function that is the pressure $p(T, \mu, \tilde{\mu})$, and the relevant relations[12]

$$\epsilon + p = sT + \rho\mu + \tilde{\rho}\tilde{\mu}, \tag{3.10}$$

$$d\epsilon = Tds + \mu d\rho + \tilde{\mu}d\tilde{\rho}. \tag{3.11}$$

Notice that here we are discussing co-dimension 1 charged planes as opposed to [8] where strings were present. This explains why the tension appears differently in (3.3) compared to [8].

### 3.1.1 Entropy current conservation and anomaly

We now want to show that the entropy current is conserved at this order in the hydrodynamic expansion. In the process, we will obtain the values of the yet undefined scalar functions. We consider the following combination of the equations of motion:

$$\Omega = u_{\nu}\partial_{\mu}T^{\mu\nu} + \mu\partial_{\mu}J^{\mu} + \tilde{\mu}u^{\mu}h^{\nu}\left(\partial_{\mu}(\star K)_{\nu} - \partial_{\nu}(\star K)_{\mu}\right). \tag{3.12}$$

This quantity can be computed using the constitutive relations (3.3-3.5) as well as the thermodynamic relations (3.10) and (3.11) but leaving $\tau, \gamma, \sigma$ and $\tilde{\sigma}$ arbitrary. We obtain

$$\begin{aligned}\Omega = &-T\partial_{\mu}(su^{\mu}) + (\tau - \tilde{\mu}\tilde{\rho})\Delta^{\mu\nu}\partial_{\mu}u_{\nu} - (\gamma - \mu\sigma)\partial_{\mu}h^{\mu} + (\gamma - \tilde{\mu}\tilde{\sigma})u^{\mu}u^{\nu}\partial_{\mu}h_{\nu}\\ &-h^{\mu}\partial_{\mu}\gamma + \mu h^{\mu}\partial_{\mu}\sigma + \tilde{\mu}h^{\mu}\partial_{\mu}\tilde{\sigma}.\end{aligned} \tag{3.13}$$

On the other hand, we can also compute $\Omega$ using the conservation equations (3.6), (3.7) and (3.8). In this case, we obtain

$$\Omega = u^{\mu}h^{\nu}F_{\mu\nu}(\sigma - a\tilde{\mu}). \tag{3.14}$$

If there were no anomaly ($a = 0$), it is trivial to see that $\tau - \tilde{\mu}\tilde{\rho} = \gamma = \sigma = \tilde{\sigma} = 0$ is the only possibility that yields a conserved entropy current which is at rest in the frame defined by $u^{\mu}$.

---

[11]Only $d-1$ dynamical equations can be derived from (3.8). The rest are constraints on the initial conditions.

[12]Note that our definition of the pressure differs from the one in [33] and [35] because we want it to be symmetric in terms of tilde and non-tilde quantities. This also explains the difference in (3.3).

This is the statement that charges must be at rest in the same frame as the energy/entropy. With the anomaly, the equation (3.14) fixes $\sigma = a\tilde{\mu}$ since it must be zero for arbitrary $F_{\mu\nu}$. For the entropy to be conserved in arbitrary flows and backgrounds, we have to impose

$$\tau = \tilde{\mu}\tilde{\rho}, \qquad \gamma = a\mu\tilde{\mu}, \qquad \sigma = a\tilde{\mu}, \qquad \tilde{\sigma} = a\mu. \qquad (3.15)$$

Allowing for the identifications described in appendix B, this agrees exactly with the results from [33] and [32].

Notice that this manifestation of the interplay between anomalies and entropy current conservation is, in some way, a simpler version of the first example discussed in [9]. There the effect appeared at first order in the derivative expansion, while here it is already present at zeroth order.

## 3.2 First order hydrodynamics

Hydrodynamics is organised as a derivative expansion. The constitutive equations (3.3-3.5) are only the zeroth-order term in this expansion. Here, we construct the next order as

$$T^{\mu\nu} = T^{\mu\nu}_{(0)} + T^{\mu\nu}_{(1)} + \dots, \qquad (3.16)$$

$$J^{\mu} = J^{\mu}_{(0)} + J^{\mu}_{(1)} + \dots, \qquad (3.17)$$

$$(\star K)^{\mu} = (\star K)^{\mu}_{(0)} + (\star K)^{\mu}_{(1)} + \dots. \qquad (3.18)$$

The first order corrections are parametrized in terms of scalar quantities called transport coefficients and dissipation appears at this order. The requirement that the entropy has to increase over time strongly constrains these corrections.

In constructing first order corrections, discrete symmetries such as charge conjugation ($C$) and parity ($P$) play an important role. Notice that, because of the anomaly, there is only one notion of charge conjugation that changes the signs of $J$ and $K$ simultaneously. In this work, we assume the charge assignments displayed in Table 1.

Table 1: Charges under discrete symmetries for 0-form symmetry

|   | $T^{\mu\nu}$ | $J^{\mu}$ | $(\star K)^{\mu}$ | $u^{\mu}$ | $h^{\mu}$ | $\epsilon, p, \tau, \gamma$ | $\rho, \mu, \sigma$ | $\tilde{\rho}, \tilde{\mu}, \tilde{\sigma}$ |
|---|---|---|---|---|---|---|---|---|
| P | + | + | + | + | + | + | + | + |
| C | + | − | − | + | + | + | − | − |

The most general corrections that we can write for the first-order terms are

$$T^{\mu\nu}_{(1)} = \delta\epsilon\, u^{\mu}u^{\nu} + \delta f \Delta^{\mu\nu} + \delta\tau h^{\mu}h^{\nu} + \ell^{(\mu}h^{\nu)} + m^{(\mu}u^{\nu)} + t^{\mu\nu}, \qquad (3.19)$$

$$J^{\mu}_{(1)} = \delta\rho\, u^{\mu} + \delta\sigma h^{\mu} + j^{\mu}, \qquad (3.20)$$

$$(\star K)^{\mu}_{(1)} = \delta\tilde{\sigma}\, u^{\mu} + \delta\tilde{\rho} h^{\mu} + k^{\mu}. \qquad (3.21)$$

In this decomposition, $l^{\mu}$, $m^{\mu}$, $j^{\mu}$ and $k^{\mu}$ are transverse vectors to both $u^{\mu}$ and $h^{\mu}$, and $t^{\mu\nu}$ is a symmetric traceless tensor. Note that in (3.19), we have not added a term $\delta\gamma\, u^{(\mu}h^{\nu)}$. This is because as explained previously, we can always boost our system in the $(u,h)$ plane to modify the value of $\gamma$. Our frame is fixed once and for all at zeroth order by choosing the entropy current to remain at rest.

In hydrodynamics, we have the freedom to change the hydrodynamical frame. This is because the fluid variables $\{u^{\mu}, h^{\mu}, \mu, \tilde{\mu}, T\}$ have no intrinsic microscopic definition out of equilibrium. The currents and the stress-energy tensor must be invariant under such redefinition.

We use the scalar redefinitions of $\mu, \tilde{\mu}$, and $T$ to set $\delta\rho = \delta\tilde{\rho} = \delta\epsilon = 0$ and the two vector redefinitions of $u^\mu$ and $h^\mu$ to set $l^\mu = m^\mu = 0$. We end up with the simpler first order expansion:

$$T_{(1)}^{\mu\nu} = \delta f \Delta^{\mu\nu} + \delta\tau h^\mu h^\nu + t^{\mu\nu}, \tag{3.22}$$

$$J_{(1)}^{\mu} = \delta\sigma h^\mu + j^\mu, \tag{3.23}$$

$$(\star K)_{(1)}^{\mu} = \delta\tilde{\sigma} u^\mu + k^\mu. \tag{3.24}$$

To proceed, we need to determine the most general form of the first order corrections $\{\delta f, \delta\tau, \delta\sigma, \delta\tilde{\sigma}, j^\mu, k^\mu, t^{\mu\nu}\}$ in terms of derivatives of fluid variables. This is done in appendix C. In any case, most possible structures do not appear as a consequence of the second law of thermodynamics to which we turn now.

The entropy current needs to be modified to first order in derivatives as

$$S^\mu = s u^\mu - \frac{1}{T} T_{(1)}^{\mu\nu} u_\nu - \frac{\mu}{T} J_{(1)}^{\mu} - \frac{\tilde{\mu}}{T} (\star K)_{(1)\nu} h^{[\nu} u^{\mu]}. \tag{3.25}$$

One can easily check that this combination is invariant under frame redefinitions as required [36]. Note that, in (3.25), we could have expected corrections coming from the anomaly as in [9]. However, this is not the case as the anomaly was already included at zeroth order.

We can now compute the divergence of this quantity. After some algebra, we obtain

$$\partial_\mu S^\mu = -T_{(1)}^{\mu\nu} \partial_\mu\left(\frac{u_\nu}{T}\right) - J_{(1)}^{\mu}\left(\partial_\mu\left(\frac{\mu}{T}\right) - \frac{u^\nu F_{\nu\mu}}{T}\right) - (\star K)_{(1)\nu} \partial_\mu\left(\frac{\tilde{\mu}}{T} h^{[\nu} u^{\mu]}\right). \tag{3.26}$$

The second law of thermodynamics implies that the right hand side of (3.26) must always be positive. Because the contributions to the divergence of the entropy current decompose is scalar, vector and tensor channels, we can impose positivity on each sector separately. This fixes completely the form the first order correction to the constitutive equations up to a number of transport coefficients. Concretely, in the tensor sector,

$$t^{\mu\nu} = -\eta\left(\Delta^{\mu\alpha}\Delta^{\nu\beta} - \frac{1}{d-2}\Delta^{\mu\nu}\Delta^{\alpha\beta}\right)\partial_{(\alpha}u_{\beta)}, \tag{3.27}$$

where $\eta$ is the shear viscosity and must be positive. The vector sector yields,

$$\begin{pmatrix} j^\mu \\ k^\mu \end{pmatrix} = -\Delta^{\mu\rho}\begin{pmatrix} \Sigma_{11} & \Sigma_{12} \\ \Sigma_{12} & \Sigma_{22} \end{pmatrix}\begin{pmatrix} \partial_\rho\left(\frac{\mu}{T}\right) - \frac{u^\sigma F_{\sigma\rho}}{T} \\ \partial_\sigma\left(\frac{\tilde{\mu}}{T} h_{[\rho} u^{\sigma]}\right) \end{pmatrix}. \tag{3.28}$$

The matrix $\Sigma$ of conductivities must be positive semi-definite implying $\Sigma_{11} \geq 0$ and $\Sigma_{11}\Sigma_{22} \geq \Sigma_{12}^2$. Onsager relations enforce that this matrix must be symmetric [36]. A small detail is that the vector structures used above contain terms that include time derivatives. This term can easily be removed by the considerations of appendix C and written in terms of other structures if one wanted to preserve the nature of the initial value problem.

In the scalar sector,

$$\begin{pmatrix} \delta f \\ \delta\tau \\ \delta\sigma \\ \delta\tilde{\sigma} \end{pmatrix} = -\begin{pmatrix} \zeta_{11} & \zeta_{12} & \zeta_{13} & \zeta_{14} \\ \zeta_{12} & \zeta_{22} & \zeta_{23} & \zeta_{24} \\ \zeta_{13} & \zeta_{23} & \zeta_{33} & \zeta_{34} \\ \zeta_{14} & \zeta_{24} & \zeta_{34} & \zeta_{44} \end{pmatrix}\begin{pmatrix} \Delta^{\mu\nu}\partial_\mu\left(\frac{u_\nu}{T}\right) \\ h^\mu h^\nu \partial_\mu\left(\frac{u_\nu}{T}\right) \\ h^\mu\left(\partial_\mu\left(\frac{\mu}{T}\right) - \frac{u^\nu F_{\nu\mu}}{T}\right) \\ h^\mu\partial_\mu\left(\frac{\tilde{\mu}}{T}\right) + \left(\frac{\tilde{\mu}}{T}\right)\Delta^{\mu\nu}\partial_\mu h_\nu \end{pmatrix}. \tag{3.29}$$

Once again, the matrix of transport coefficients in equation (3.29) has to be symmetric due to Onsager relations on mixed correlation functions [36] as well as positive definite. This matrix contains terms such as bulk viscosities and components of the conductivity.

All in all, we have fourteen transport coefficients that are split as $1 + 3 + 10 = 14$ in the tensor, vector and scalar sectors respectively. This completely agrees with [33].

# 4 Generalization to higher-form superfluids

Using the technology of the previous section one can easily build the equivalent hydrodynamic theories for systems enjoying a $p$-form abelian symmetry $U(1)^{(p)}$ that is spontaneously broken. All the physics is, in this case, contained in an anomalous emergent $U(1)^{(d-p-2)}$ symmetry. We sketch an example of this construction for $p = 1$ and $d = 4$; generalizations to other cases are straightforward. This particular system describes the hydrodynamic behavior of Quantum Electrodynamics (QED) at energy scales below the electron mass. The Goldstone mode is none other than the (partially screened) photon.

The results obtained here match the construction in [37] in terms of an effective action.

## 4.1 1-form superfluid hydrodynamics

Consider a system with a $U(1)^{(1)} \times U(1)^{(1)}$ symmetry in $d = 4$ in the presence of a background two-form gauge field $B$ that couples to one of the $U(1)$ currents (which we call magnetic, keeping the QED example in mind). The conservation equations for this system read

$$\partial_\mu T^{\mu\nu} = H^{\nu\alpha\beta} J_{\alpha\beta}, \tag{4.1}$$

$$\partial_\mu J^{\mu\nu} = 0, \tag{4.2}$$

$$\partial_\mu K^{\mu\nu} = -\frac{a}{3}\epsilon^{\nu\alpha\beta\gamma} H_{\alpha\beta\gamma}, \tag{4.3}$$

where

$$H_{\alpha\beta\gamma} = \partial_\alpha B_{\beta\gamma} - \partial_\beta B_{\alpha\gamma} + \partial_\gamma B_{\alpha\beta}, \tag{4.4}$$

and $B_{\mu\nu}$ is a two-form gauge potential. Notice that this system, in a non-trivial state, possesses no continuous space-time symmetries, as the magnetic and electric field can point in arbitrary directions. In these conditions a new situation arises: the charges, even in equilibrium, need not be collinear with the chemical potentials. As all space-time symmetries are broken we can pick a basis of orthonormal vectors:

$$u_\mu u^\mu = -1, \qquad h_\mu h^\mu = e_\mu e^\mu = 1, \qquad u^\mu h_\mu = u^\mu e_\mu = h^\mu e_\mu = 0, \tag{4.5}$$

and write

$$\mu^\mu = \mu h^\mu, \qquad \tilde{\mu}^\mu = \tilde{\mu}_\parallel h^\mu + \tilde{\mu}_\perp e^\mu. \tag{4.6}$$

Here, $u^\mu$ is the fluid velocity as in conventional hydrodynamics, $h^\mu$ indicates the direction of the magnetic chemical potential related to $J$ while the electrical analog quantity related to $K$ is contained in the $(h, e)$ plane. The most general constitutive relations for the currents are

$$J^{\mu\nu} = \rho u^{[\mu}h^{\nu]} + \rho_\times u^{[\mu}e^{\nu]} + \sigma_\parallel \epsilon^{\mu\nu\rho\sigma} u_\rho h_\sigma + \sigma_\perp \epsilon^{\mu\nu\rho\sigma} u_\rho e_\sigma, \tag{4.7}$$

$$K^{\mu\nu} = \tilde{\rho}_\parallel u^{[\mu}h^{\nu]} + \tilde{\rho}_\perp u^{[\mu}e^{\nu]} + \tilde{\sigma} \epsilon^{\mu\nu\rho\sigma} u_\rho h_\sigma + \tilde{\sigma}_\times \epsilon^{\mu\nu\rho\sigma} u_\rho e_\sigma. \tag{4.8}$$

Because charges and chemical potentials don't need to be aligned, we cannot remove any of the structures above. Notice that equations (4.7-4.8) allow the inclusion of a parity odd structure. This follows from the existence of a parity odd scalar in this system. In QED this is the familiar scalar product between the electric and magnetic field. In Table 2 we display our conventions for charge conjugation ($C$), which as in the previous section reverses the sign of both $J$ and $K$, and parity ($P$), appropriate for QED.

With these charges under discrete symmetries, we can write the constitutive equation for the stress-energy tensor

$$T^{\mu\nu} = (\epsilon + p)u^\mu u^\nu + p\,\eta^{\mu\nu} - \tau\,h^\mu h^\nu - \tilde{\tau}\,e^\mu e^\nu - \varphi\,h^{(\mu}e^{\nu)} - \gamma\,\epsilon^{(\mu\alpha\beta\gamma}u_\alpha h_\beta e_\gamma u^{\nu)}, \tag{4.9}$$

Table 2: Charges under discrete symmetries for 1-form symmetry

|   | $T^{\mu\nu}$ | $J^{\mu\nu}$ | $K^{\mu\nu}$ | $u^\mu$ | $h^\mu$ | $e^\mu$ | $\epsilon, p, \tau, \tilde\tau, \gamma$ | $\varphi$ | $\mu, \rho, \tilde\sigma$ | $\tilde\mu_\parallel, \tilde\rho_\parallel, \sigma_\parallel$ | $\tilde\mu_\perp, \tilde\rho_\perp, \sigma_\perp$ | $\rho_\times, \tilde\sigma_\times$ |
|---|---|---|---|---|---|---|---|---|---|---|---|---|
| P | + | − | + | + | − | + | + | − | + | − | + | − |
| C | + | − | − | + | − | − | + | + | + | + | + | + |

where $\epsilon$ is the energy density, $p$ is the pressure and $\tau, \tilde\tau, \varphi$ parameterize the stress tensor of magnetic and electric strings[13]. The $\gamma$ term contains the effect of the anomaly. While the symmetries allow another term quadratic in $\epsilon^{\mu\nu\rho\sigma}$, this term in not linearly independent from the $\eta^{\mu\nu}$ term.

The thermodynamics is completely specified by an equation of state and the relevant thermodynamic relations are

$$\epsilon + p = sT + \rho_\mu \mu^\mu + \tilde\rho_\mu \tilde\mu^\mu, \tag{4.10}$$

$$d\epsilon = Tds + \mu_\mu d\rho^\mu + \tilde\mu_\mu d\tilde\rho^\mu, \tag{4.11}$$

where

$$\rho^\mu = \rho h^\mu + \rho_\times e^\mu, \qquad \tilde\rho^\mu = \tilde\rho_\parallel h^\mu + \tilde\rho_\perp e^\mu. \tag{4.12}$$

In a less covariant, but more transparent notation these equation can be rewritten as:

$$\epsilon + p = sT + \rho\mu + \tilde\rho_\parallel \tilde\mu_\parallel + \tilde\rho_\perp \tilde\mu_\perp, \tag{4.13}$$

$$d\epsilon = Tds + \mu d\rho + \mu\rho_\times h_\mu de^\mu + \tilde\mu_\parallel d\tilde\rho_\parallel + \tilde\mu_\parallel \tilde\rho_\perp h_\mu de^\mu + \tilde\mu_\perp d\tilde\rho_\perp + \tilde\mu_\perp \tilde\rho_\parallel e_\mu dh^\mu. \tag{4.14}$$

Provided we can use the conservation of the entropy current and the anomaly argument from the previous section to fix uniquely $\sigma_\parallel, \sigma_\perp, \tilde\sigma, \tilde\sigma_\times, \tau, \tilde\tau, \varphi, \gamma, \rho_\times$, the above is a closed system of equations. In total, there are ten hydrodynamic variables in $\mu, \tilde\mu_\parallel, \tilde\mu_\perp, T, u^\mu, h^\mu, e^\mu$ and twelve equations of motion of which two are constraints, making the system closed.

Notice that one of the densities, which we choose to be $\rho_\times$ needs to be fixed. This is consistent with the fact that there are (due to rotational symmetry) only 3 chemical potentials $(\mu, \tilde\mu_\parallel, \tilde\mu_\perp)$ as, covariantly, the pressure $p$ can only be a function of $(\mu \cdot \mu, \tilde\mu \cdot \tilde\mu, \mu \cdot \tilde\mu)$. As a consequence, only 3 densities can be independent. This is equivalent to the final expressions[14]

$$\epsilon + p = sT + \rho\mu + \tilde\rho_\parallel \tilde\mu_\parallel + \tilde\rho_\perp \tilde\mu_\perp, \tag{4.15}$$

$$d\epsilon = Tds + \mu d\rho + \tilde\mu_\parallel d\tilde\rho_\parallel + \tilde\mu_\perp d\tilde\rho_\perp, \tag{4.16}$$

$$dp = sdT + \rho d\mu + \tilde\rho_\parallel d\tilde\mu_\parallel + \tilde\rho_\perp d\tilde\mu_\perp, \tag{4.17}$$

$$\rho_\times = \frac{\tilde\mu_\perp \tilde\rho_\parallel - \tilde\mu_\parallel \tilde\rho_\perp}{\mu}. \tag{4.18}$$

We now proceed as with the 0-form case and demand the entropy current to be conserved. Consider

$$\Omega = u_\nu \partial_\mu T^{\mu\nu} + \mu h_\nu \partial_\mu J^{\mu\nu} + \tilde\mu_\parallel h_\nu \partial_\mu K^{\mu\nu} + \tilde\mu_\perp e_\nu \partial_\mu K^{\mu\nu}. \tag{4.19}$$

This quantity can be computed from the conservation equations (4.1-4.3) to give

$$\Omega = H^{\nu\alpha\beta} \left[ u_\nu \epsilon_{\alpha\beta\rho\sigma} u^\rho \left( \sigma_\parallel h^\sigma + \sigma_\perp e^\sigma \right) - \frac{a}{3} \epsilon_{\lambda\nu\alpha\beta} \left( \tilde\mu_\parallel h^\lambda + \tilde\mu_\perp e^\lambda \right) \right]. \tag{4.20}$$

---

[13]In principle, techniques similar to those displayed in appendix A can be used to find the values of $\tau, \tilde\tau, \varphi$. In this case one needs to consider more general volume preserving deformations of the fluid element, not present for the 0-form case. These are important in the theory of elasticity and have been considered in a modern setup recently in [38].

[14]This result also follows direct from entropy conservation as explained below.

This must vanish for arbitrary $H^{\nu\alpha\beta}$ in order for the entropy to be conserved. On the other hand, using constitutive relations (4.7), (4.8) and (4.9) we must obtain $\Omega = -T\partial_\mu(su^\mu)$. Both these conditions can only be satisfied provided

$$\tau = \mu\rho + \tilde{\mu}_\parallel \tilde{\rho}_\parallel,, \qquad\qquad \tilde{\tau} = \tilde{\mu}_\perp \tilde{\rho}_\perp, \qquad\qquad (4.21)$$

$$\varphi = \tilde{\mu}_\perp \tilde{\rho}_\parallel, \qquad\qquad \rho_\times = \frac{\tilde{\mu}_\perp \rho_\parallel - \tilde{\mu}_\parallel \tilde{\rho}_\perp}{\mu}, \qquad\qquad (4.22)$$

$$\sigma_\parallel = a\tilde{\mu}_\parallel, \qquad\qquad \sigma_\perp = a\tilde{\mu}_\perp, \qquad\qquad (4.23)$$

$$\tilde{\sigma} = -a\mu, \qquad\qquad \tilde{\sigma}_\times = 0, \qquad\qquad (4.24)$$

$$\gamma = a\mu\tilde{\mu}_\perp. \qquad\qquad (4.25)$$

This results in a hydrodynamic system equivalent to the one presented in [37], if one considers the set of identifications presented in appendix B.

With this information, it is straightforward to follow the standard procedure outlined in the previous section and construct higher order corrections to the constitutive relations in the derivative expansion. We will not do this in this present work, but refer the reader instead to [37] for the general structure of these corrections, albeit in a different formalism.

# 5 Outlook

In this work, we have shown that the masslessness of bosons coming from spontaneous breaking of abelian symmetries (superfluids, photons, etc.) can be interpreted as being protected by an anomaly, analogously to the masslessness of fermions. This observation was upgraded into a fundamental principle by reversing the logic and classifying certain phases of matter by their (higher-form) symmetries and their anomalies without reference to how the symmetries are realized. As an example of this program we have presented constructions of 0-form and 1-form superfluids in a systematic fashion in a formalism that puts the Josephson relation on equal footing with the other conservation equations.

It is tempting to explore the consequences of this new paradigm. For example, are all gapless phases protected by anomalies? Goldstones for non-abelian symmetries are parametrized by an element of a coset $g \in G/H$. The natural generalization of the current $\star K = d\phi$ to this situation is the Maurer-Cartan form $\star K \equiv g^{-1}dg$. This current fails to be conserved but instead satisfies the Cartan structure equation

$$d \star K = -(\star K) \wedge (\star K). \qquad\qquad (5.1)$$

Since sigma models are IR-free in $d \geq 3$, the theory abelianizes at low energies where one can neglect the non-linear term in (5.1). A mixed abelian anomaly of the form (1.4) can therefore also be said to protect the massless modes of this non-abelian theory.

There are other generalizations to non-abelian groups that are possible in certain circumstances. Although higher-form symmetries are always abelian (because the objects counting higher-form charges have enough codimensions to be swapped nonviolently), the 0-form symmetry can be non-abelian. The anomaly (1.4) is canceled by inflow from a bulk term $\int B \wedge F$. One natural generalization is for the bulk term to be replaced with $\int B \wedge \operatorname{Tr} F^n$. For $n$ even, a theory with an anomaly of this form is given by the $G_L \times G_R / G_{\text{diag}}$ sigma-model. Focusing on $n = 2$ for concreteness, this theory can have a closed Wess-Zumino (WZ) 3-form $\omega^{(3)}$ (which can be used to add a WZ term to the theory in 2 dimensions), whose closedness is spoiled if a certain subgroup $F \subset G_L \times G_R$ is gauged [39] – a simple example is $F = G_L$. The best improvement of $\omega^{(3)}$ that one can construct then satisfies

$$d\omega^{(3)} \propto \operatorname{Tr}\left(F_L^2 - F_R^2\right). \qquad\qquad (5.2)$$

The theory therefore contains an anomalous $U(1)^{(d-4)}$ symmetry carried by the $(d-3)$-from current $\star K \equiv \omega^{(3)}$ – this is the skyrmion number current [40][15]. See Ref. [41] for further obstructions to gauging WZ terms. Interesting generalizations that can accommodate a non-abelian structure in a more fundamental fashion, such as connections to 2-groups [18], should be pursued. We leave this for future work.

The ideas discussed here are in line with recent progress in incorporating higher form symmetries in hydrodynamics [7, 8, 37, 38, 42–46] and we expect to see further developments in this area.

The identification of higher form symmetries in this paper also revealed the fact that the BKT transition is a regular Landau transition between two phases with different symmetries (and anomalies). One can then ask: which phases are truly non-Landau, after generalized symmetries (continuous and discrete) and their anomalies are taken into account? Certain fractional quantum hall phases[16] can also be distinguished by the symmetries of their effective Chern-Simons descriptions. A discrete higher-form symmetry similarly distinguishes both phases of the Ising model, obviating the need to specify whether the symmetries are spontaneously broken or not. It is important to understand if and under which conditions the Landau paradigm effectively fails.

It would also be of interest to use this new point of view to shed light on the traditional treatment of superfluids within the gauge/gravity duality [35, 47, 48]. The proper treatment of higher form symmetries within holography requires the inclusion of bulk Chern-Simons terms and a careful consideration of the boundary conditions in some cases [49]. A second look at this system might provide a holographic version of the anomaly inflow mechanism described in section 1.1, giving a clearer connection to the study of topological phases.

# Acknowledgments

We thank Blaise Goutéraux, Nabil Iqbal, Anton Kapustin, John McGreevy, Nick Poovuttikul, Brandon Rayhaun, Dam T. Son, John Stout and David Tong for inspiring discussions. L.D. would like to acknowledge the hospitality of the organizers of the Amsterdam String Theory Workshop where some of this work was conducted. L.D. is partially supported by DOE award DE-SC0018134 (Sean Hartnoll). D.H. and G.M. are supported in part by the ERC Starting Grant GENGEOHOL.

# A  Thermodynamic argument to fix $\tau$

Consider a system that contains surfaces of area A running perpendicular to lines of length L, with an associated tension $\tau$ and a conserved charge $\tilde{Q}$ given by the number of planes through the line. The variation of the internal energy for this system is

$$dU = TdS - pdV + \tau LdA + \tilde{\mu}Ad\tilde{Q}. \tag{A.1}$$

Now, since $\tilde{Q}$ is defined by a line integral, it is given by $\tilde{Q} = \tilde{\rho}L$. Consider a Legendre transform to the Landau grand potential:

$$\Phi = U - TS - \tilde{\mu}A\tilde{Q}, \tag{A.2}$$

$$d\Phi = -sVdT - pdV - \tilde{\rho}Vd\tilde{\mu} + (\tau - \tilde{\rho}\tilde{\mu})LdA, \tag{A.3}$$

---

[15]Note that in the context of chiral symmetry breaking, this symmetry is not emergent but carries the baryon number $U(1)$ present in the UV, as required by anomaly matching.

[16]Interestingly, the simpler case of integer quantum hall states is more resistant to the Landau paradigm treatment. D.H. thanks Anton Kapustin and David Tong for discussions on this issue.

where $s$ is the entropy density. This quantity is naturally calculated by the on-shell action; we thus expect this to scale with the volume. This scaling is spoiled unless $\tau = \tilde{\mu}\tilde{\rho}$.

## B Conversion between conventions

We provide here the map between our results and the well established zeroth order superfluid hydrodynamics results from [33]. They are given, in the form (theirs = ours) by

$$\sqrt{\xi^2 + a^2\mu^2} = \tilde{\rho}\,, \qquad f_s = \tilde{\rho}/\tilde{\mu}\,, \qquad \xi^\mu = a\mu u^\mu + \tilde{\rho}h^\mu\,, \tag{B.1}$$
$$n + f_s a^2\mu = \rho\,, \qquad \epsilon + f_s a^2\mu^2 = \epsilon\,, \qquad P = P - \tilde{\mu}\tilde{\rho}\,. \tag{B.2}$$

Below we list the map between the conventions in [37] and our results from section 4. As before, they are given in the form (theirs = ours)

$$\bar{\zeta}^\mu = -\tilde{\rho}_\parallel^\mu h^\mu - \tilde{\rho}_\perp e^\mu\,, \qquad\qquad\qquad \zeta^\mu = -\mu h^\mu\,, \tag{B.3}$$
$$\bar{q} = -\frac{\tilde{\mu}_\perp}{\tilde{\rho}_\perp}\,, \qquad q_\times = -\frac{\tilde{\mu}_\parallel}{\mu} + \frac{\tilde{\mu}_\perp\tilde{\rho}_\parallel}{\tilde{\rho}_\perp\mu}\,, \qquad q = \frac{\rho}{\mu} + \frac{\tilde{\mu}_\parallel\tilde{\rho}_\parallel}{\mu^2} - \frac{\tilde{\mu}_\perp\tilde{\rho}_\parallel^2}{\tilde{\rho}_\perp\mu^2}\,, \tag{B.4}$$
$$\epsilon_{them} = \epsilon_{us}\,, \qquad p_{them} = p_{us}\,, \qquad -1 = a_{us}\,. \tag{B.5}$$

## C First order tensor structures in 0-form superfluids

**Scalars**

We are looking for all the scalars that we can construct out of $\{T, \mu, \tilde{\mu}, u^\mu, h^\mu\}$ with exactly one derivative[17]. Let us start by listing all linearly independent scalars:

$$u^\lambda \partial_\lambda T\,, \qquad\qquad\qquad h^\lambda \partial_\lambda T\,, \tag{C.1}$$
$$u^\lambda \partial_\lambda \frac{\mu}{T}\,, \qquad\qquad\qquad h^\lambda \partial_\lambda \frac{\mu}{T}\,, \tag{C.2}$$
$$u^\lambda \partial_\lambda \frac{\tilde{\mu}}{T}\,, \qquad\qquad\qquad h^\lambda \partial_\lambda \frac{\tilde{\mu}}{T}\,, \tag{C.3}$$
$$\Delta^{\mu\nu} \partial_\mu u_\nu\,, \qquad\qquad\qquad \Delta^{\mu\nu} \partial_\mu h_\nu\,, \tag{C.4}$$
$$h^\mu h^\nu \partial_\mu u_\nu\,, \qquad\qquad\qquad u^\mu h^\nu \partial_\mu u_\nu\,. \tag{C.5}$$

Now, we can also use the conservation equations

$$\partial_\mu J^\mu = 0\,, \qquad \partial_{[\mu}(\star K)_{\nu]} = 0\,,, \qquad \partial_\mu T^{\mu\nu} = 0\,, \tag{C.6}$$

with which we can build four scalar equations, namely

$$\partial_\mu J^\mu = 0\,, \tag{C.7}$$
$$u^{[\mu}h^{\nu]}\partial_{[\mu}(\star K)_{\nu]} = 0\,, \tag{C.8}$$
$$u_\nu \partial_\mu T^{\mu\nu} = 0 \tag{C.9}$$
$$h_\nu \partial_\mu T^{\mu\nu} = 0\,. \tag{C.10}$$

---

[17]In this section we turn off the background gauge field $A$.

We use these equations to zeroth order to further remove terms containing time derivatives. This preserves the nature of the initial value problem. This way we remove $u^\lambda \partial_\lambda T$, $u^\mu h^\nu \partial_\mu u_\nu$, $u^\lambda \partial_\lambda \mu$ and $u^\lambda \partial_\lambda \tilde{\mu}$. Finally, we take our set of independent scalars to be

$$h^\lambda \partial_\lambda \frac{\mu}{T}\,, \qquad\qquad\qquad h^\lambda \partial_\lambda \frac{\tilde{\mu}}{T}\,, \tag{C.11}$$

$$h^\mu h^\nu \partial_\mu u_\nu\,, \qquad\qquad\qquad h^\lambda \partial_\lambda T\,, \tag{C.12}$$

$$\Delta^{\mu\nu}\partial_\mu u_\nu\,, \qquad\qquad\qquad \Delta^{\mu\nu}\partial_\mu h_\nu\,, \tag{C.13}$$

where the first line is charge odd and the two remaining lines are charge even. Of these structures, only four will be allowed by the second law of thermodynamics (3.29).

**Vectors**

The transverse vector conservation equations are

$$\Delta_{\rho\alpha}\partial_\mu T^{\mu\alpha} = 0\,, \tag{C.14}$$

$$\Delta^{\mu\alpha} u_\alpha \partial_{[\mu}(\star K)_{\nu]} = 0\,, \tag{C.15}$$

$$\Delta^{\mu\alpha} h_\alpha \partial_{[\mu}(\star K)_{\nu]} = 0\,. \tag{C.16}$$

All the transverse vector quantities that we can consider are

$$\Delta^{\mu\nu}\partial_\nu T\,, \qquad\qquad\qquad \Delta^{\mu\nu}\partial_\nu \frac{\mu}{T}\,, \tag{C.17}$$

$$\Delta^{\mu\nu}\partial_\nu \frac{\tilde{\mu}}{T}\,, \qquad\qquad\qquad \Delta^{\mu\nu} h^\lambda \partial_\nu u_\lambda\,, \tag{C.18}$$

$$\cancel{\Delta^{\mu\nu} u^\lambda \partial_\lambda u_\nu}\,, \qquad\qquad\qquad \Delta^{\mu\nu} h^\lambda \partial_\lambda u_\nu\,, \tag{C.19}$$

$$\cancel{\Delta^{\mu\nu} u^\lambda \partial_\lambda h_\nu}\,, \qquad\qquad\qquad \cancel{\Delta^{\mu\nu} h^\lambda \partial_\lambda h_\nu}\,, \tag{C.20}$$

where we have used (C.14-C.16) to get rid of the three last components. Only two structures are allowed by the second law of thermodynamics (3.28).

**Tensors**

In this case there are no transverse symmetric tensor equations. The transverse traceless symmetric tensors are given by

$$\sigma^{\mu\nu} = \left(\Delta^{\mu\alpha}\Delta^{\nu\beta} - \frac{1}{d-2}\Delta^{\mu\nu}\Delta^{\alpha\beta}\right)\partial_{(\alpha} u_{\beta)}\,, \tag{C.21}$$

$$\zeta^{\mu\nu} = \left(\Delta^{\mu\alpha}\Delta^{\nu\beta} - \frac{1}{d-2}\Delta^{\mu\nu}\Delta^{\alpha\beta}\right)\partial_{(\alpha} h_{\beta)}\,. \tag{C.22}$$

Only one structure is allowed by the second law of therodynamics (3.27).

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
