# Peer review of "Superfluids as Higher-form Anomalies"

_SciPost Physics, doi:SciPost Phys. 8, 047 (2020)_

## Round 1 · Referee Report · Anonymous (Referee 1) · 2019-10-15

Strengths

  1. new perspective on superfluid hydrodynamics from higher-form symmetries
  2. mixed anomalies involving higher-form symmetries predict/protect certain massless modes in the EFT (such as the goldstone field in superfluid)
  3. streamlined understanding of the usual hydrodynamic equations by including higher-form symmetry currents

Weaknesses

See requested changes

Report

The authors provide a new perspective on conventional superfluid hydrodynamics by identifying an emergent higher-form symmetry in the EFT: a (d-2)-from symmetry, and analyzing its implications. In particular, due to a mixed anomaly with the ordinary U(1) 0-form symmetry, they explained nicely using just the correlators of the conserved currents that a goldstone mode for the U(1) 0-form symmetry must be present (without assuming SSB). They went on to show that by incorporating (anomalous) conservation and constituent relations for the higher form symmetry, one recover the usual hydrodynamic equations for the ordinary superfluid in an elegant and uniform fashion. They also discussed the 1-form superfluid case which is relevant for the low energy phase of 4d QED. The paper is well-structured and nicely-written. It presents a fruitful dialogue between generalized symmetries, anomalies and hydrodynamics. I recommend this paper for publication.

Requested changes

The current as written in (1.3) is not conserved. The actual (d-1)-form current is *d\phi. But this is not invariant under the 0-from U(1) gauge transformation. What the authors wrote is the gauge invariant improvement whose non-conservation leads to the anomaly. It maybe worthwhile to explain this.

---

## Round 1 · Referee Report · Anonymous (Referee 2) · 2020-1-22

Strengths

  1. Elegant and systematic formulation of superfluid hydrodynamics.
  2. Provide several comments with physical insights.
  3. There are several potential interesting extensions.

Weaknesses

I could not find any.

Report

The paper develops a novel formalism to describe superfluid hydrodynamics in the language of generalized global symmetries and anomalies. A crucial ingredient is an emergent (d-2)-form symmetry associated with the conservation of superfluid winding planes. This symmetry has a mixed anomaly with the standard 0-form symmetry. Through a current algebra argument, the anomaly is shown to imply the existence of the Goldstone mode. The Josephson relation is stated in terms of the constitutive relation for the emergent (d-1)-form current and is naturally written in terms of a systematic derivative expansion. In transitioning to the normal fluid phase, the (d-2)-form symmetry is explicitly broken by the proliferation of vortices. The formalism can be generalized to higher-form superfluids in any dimension. As an additional example, the authors apply the formalism to QED in 4 dimensions at scales below the electron mass, which can be viewed as a 1-form superfluid. In this case, the emergent symmetry is the electric 1-form symmetry. The paper has a clear structure and contains results which can lead to further investigation on this topic. I therefore recommend the paper for publication.

Requested changes

None.

---

## Round 2 · Author Response

We thank both referees for their positive feedback, and referee 1 for their suggested change. We clarified this point in footnote 2.

---

## Round 2 · List of Changes

- Added footnote 2, explaining that the conserved (d-1)-form current is *d\phi, but is not gauge invariant.

- Sharpened Eq. (1.24), which now not only shows that the currents are not aligned, but also provides an expression for the normal density of a relativistic superfluid at low temperatures.

Resubmission 1908.06977v2 on 19 February 2020
Submission 1908.06977v1 on 15 September 2019

---

## Editorial Decision

published